# Too Hot to Ignore: The Escalating Health Impact of Heatwaves in Brazil

**DOI:** 10.3390/ijerph22091451

**Published:** 2025-09-18

**Authors:** Jessica M. Neves, Klauss K. S. Garcia, Beatriz F. A. Oliveira, Marco A. Horta

**Affiliations:** 1Laboratory of Hantavirus and Rickettsias/BSL-3 Facility, Instituto Oswaldo Cruz, Fundação Oswaldo Cruz (IOC-Fiocruz), Rio de Janeiro 21040-900, RJ, Brazil; milenamoura80@hotmail.com; 2Department of Infectious Disease Epidemiology and International Health, Faculty of Epidemiology and Populational Health, The London School of Hygiene and Tropical Medicine, Keppel Street, London WC1E 7HT, UK; 3Health and Surveillance Secretariat, Ministry of Health, Brasilia 70719-040, DF, Brazil; 4Piauí Regional Office, Fundação Oswaldo Cruz (Fiocruz), Teresina 64000-128, PI, Brazil; beatriz.oliveira@fiocruz.br

**Keywords:** Brazil, climate change, climate resilience, health impacts, heatwaves, public health

## Abstract

Heatwaves (HWs) are becoming more frequent and severe, posing a significant threat to human health. Studies have shown that extreme heat, whether as incremental temperature increases or prolonged HWs, is associated with an increased risk of heat-related illnesses. However, there is still limited understanding of how these impacts unfold in Brazil, given its unique social, environmental, and health-system contexts. I this perspective article, we explore the effects of HWs on human physiology, examine the social and biological factors that contribute to heat stress, and identify vulnerable populations at disproportionate risk. We also discuss the potential consequences of extreme heat in other aspects of society, such as agriculture and energy, and assess the challenges of strengthening resilience in Brazil’s health sector. Our key contribution are to make visible the hidden burden of heat-related mortality, to examine how fragmented governance constrains the adaptive capacity of Brazil’s health sector, and to reflect on pathways to strengthen resilience to heatwaves.

## 1. Context

Heatwaves (HWs) are prolonged periods of excessively hot weather, often accompanied by high humidity levels. Their characteristics vary by region, and no single definition exists. A common approach considers at least two to three consecutive days above long-term climatological thresholds, typically calculated over 30 years [1,2,3]. Such temperature-based metrics may be insufficient for a country as diverse as Brazil, spanning the humid Amazon, the semi-arid Northeast, and the temperate South. Evidence from Brazil, including studies in Rio de Janeiro and the Amazon, has applied relative thresholds (90th–99th percentiles of daily mean temperature for ≥2 days) and shows that mortality risk rises with heatwave intensity, especially among older adults and women [4,5]. Effective assessment may require multiple exposure indicators that capture humidity, local acclimatization, and measurable health impacts. The lack of standardized criteria hampers timely warnings and limits comparability across regions.

Brazil has experienced several HWs in recent years. The number of HWs increased from 1970 to 2020 in 14 Brazilian capitals, particularly for those in low-latitude regions [6], with a significant increase in the northern region over the past two decades (the 2000s and 2010s). The summer of 2024/2025 was the hottest in Brazil since 1961, with an average observed temperature of 26.2 °C, which was 0.73 °C above the historical average for the period from 1991 to 2020 [7]. This period was influenced by the El Niño phenomenon, which amplified warming across South America. However, other large-scale processes, such as atmospheric blocking patterns and Atlantic Sea surface temperature anomalies, also influenced heatwave dynamics in Brazil [8,9]. Against this background, the 2024 HW events were particularly severe, as they coincided with intense air pollution from forest fires in the Brazilian Amazon, creating critical episodes and heightening health risks for vulnerable populations [10,11].

HWs significantly threaten human health, causing heat-related illnesses and exacerbating pre-existing conditions. These events can lead to symptoms such as heat cramps;’ exhaustion; and, in severe cases, heatstroke. The combination of high temperature and humidity can hinder the body’s ability to cool itself through sweating, leading to heat stress and potentially life-threatening conditions. Older adults, infants, and individuals with chronic health conditions are at higher risk of heat-related mortality [12]. As global temperatures rise, the burden of heat-related morbidity and mortality is increasing. A systematic review found that for every 1 °C increase in temperature, heat illness morbidity and mortality increased by 18% and 35%, respectively [13].

Understanding the health effects of extreme heat is essential for developing effective strategies to protect vulnerable populations. This is particularly urgent in Brazil, where approximately 70% of the population relies exclusively on the public Unified Health System (SUS). The SUS is Brazil’s universal public health system, established by law to guarantee health as a right and funded through general taxation, providing free access to preventive, primary, hospital, and specialized care. In addition to triggering acute health outcomes, HWs increase the burden on healthcare services by aggravating chronic conditions and deepening social vulnerabilities. In a country marked by sharp climatic and socioeconomic disparities, these events expose systemic weaknesses and highlight the urgent need for integrated and equitable climate health responses.

Despite growing evidence on health impacts of heat, critical gaps remain in the Brazilian context. Heat-related mortality is largely invisible in official statistics, governance is fragmented, and current responses fail to adequately address the compounded risks of extreme heat, social inequalities, and environmental stressors. This opinion paper addresses this gap by discussing the structural challenges that limit the adaptive capacity of the health sector in Brazil and by reflecting on how these limitations undermine resilience to heatwaves.

## 2. Understanding the Physiological Response and Health Vulnerabilities to Extreme Heat

Prolonged exposure to extreme heat impairs the body’s ability to regulate internal temperature, increasing the risk of heat-related illnesses, such as exhaustion and heatstroke [14,15,16]. While healthy individuals rely on physiological (e.g., sweating and vasodilation) and behavioral mechanisms to regulate body temperature [17], these can be overwhelmed during intense or sustained HWs. When heat production exceeds dissipation, the core temperature increases, potentially leading to hyperthermia (an abnormally high body temperature caused by failed thermoregulation); heat exhaustion; or, in severe cases, heat stroke, requiring immediate medical attention.

Certain populations are disproportionately affected. Older adults experience age-related declines in thermoregulation, including reduced sweating and cardiovascular adaptation, which makes them more vulnerable to heat stress [12]. Children—especially infants—are at risk because of immature heat regulation and their reliance on adults to avoid exposure [18]. Individuals living with chronic conditions such as cardiovascular or metabolic diseases may have impaired heat responses that are further worsened by commonly prescribed medications [19]. Individuals with obesity are also at increased risk due to reduced heat tolerance associated with impaired heat dissipation, cardiovascular strain, and limited aerobic capacity [13,20].

The health effects of extreme heat are not limited to those of increasing temperatures. They are also shaped by unequal living conditions that influence both exposure and resilience. Individuals living in informal settlements or urban heat islands, often without access to air conditioning or green spaces, experience heightened exposure and reduced adaptive capacity [21]. Informal workers are especially vulnerable because they frequently perform outdoor labor under unsafe thermal conditions with little protection, rest, or hydration. In many urban areas, the lack of green infrastructure further compounds these risks for populations that already face systemic disadvantages [21].

## 3. Intersecting Health Risks: Chronic Diseases, Social Inequalities, and Compounding Environmental Stressors

Extreme heat is still a neglected threat to public health in Brazil. Recent studies have documented an increase in heat-related cardiovascular and respiratory mortality, with disproportionate effects on women, older adults, Black and Brown populations, and individuals with lower educational levels [4,6]. In addition, high temperatures have been linked to an increased risk of stroke and cancer-related deaths in international studies, suggesting that extreme heat may worsen pre-existing health conditions, including those in cancer patients [22,23]. In Brazil, evidence from metropolitan regions has shown excessive mortality from cancer-related mortality during periods of extreme heat exposure [6].

In addition to physical health, extreme heat poses serious risks to mental health, human behavior, and productivity. Systematic reviews link high temperatures to stress, anxiety, and sleep disturbances [24,25,26]. These effects are particularly pronounced among older adults and individuals with preexisting mental health conditions, who may be especially vulnerable to the psychological impacts of HWs.

Excessive heat can impair cognitive and physical performance. Fatigue resulting from heat stress may lead to workplace errors, thereby increasing the risk of work-related injury [27]. In rural areas of northeastern Brazil, evidence suggests that heavy labor would need to be interrupted for approximately 54% of the workday under safe exposure standards, with moderate and light activities also requiring frequent breaks [28]. Furthermore, heat exposure has been linked to rising rates of violence, including domestic violence [29,30,31], and suicide [32]. Evidence from Brazil also shows that higher temperatures are associated with increased homicidal deaths [33], while other studies link adverse climate shocks, such as droughts, to higher violent crime rates in rural municipalities [34]. The heat–suicide mortality risk has also been reported in metropolitan areas of Brazil, where women are disproportionately affected [32].

These health risks are exacerbated by deep social inequalities. HWs disproportionately affect socially vulnerable populations in Brazil, owing to inadequate urban infrastructure, limited access to healthcare, and poor living conditions. Vulnerable groups such as those in informal settlements, remote Amazonian communities, and indigenous territories often lack access to cooling systems, potable water, and public health services, making them more susceptible to heat-related illnesses and fatalities. Social inequalities further intensify these risks, leaving marginalized populations without the resources needed to adapt and protect themselves [6,35,36].

Moreover, HW events in Brazil often occur alongside concurrent environmental stressors within an already strained healthcare system. The country faces a complex epidemiological landscape marked by the (re)emergence of mosquito-borne diseases and a growing burden of mortality due to cardiovascular diseases, cancer, and external causes. These conditions amplify the adverse health effects of extreme heat. Notably, during the 2024 HW episodes, the Brazilian Amazon experienced severe wildfires, largely associated with deforestation and agricultural burning, that released high concentrations of fine particulate matter (PM_2.5_) and ground-level ozone. Combined exposure to extreme heat and air pollutants is known to act synergistically, increasing the risk of acute cardiovascular events, such as myocardial infarction and ischemic stroke [37,38]. In Amazonian cities, as well as in remote areas, these extreme temperatures coincided with water scarcity and intense smoke from wildfires, creating overlapping stressors that plausibly magnify respiratory and cardiovascular risks and deepen existing social vulnerabilities.

## 4. The Invisible Burden of Heat and the Challenge of Estimating Unrecorded Mortality

In the Brazilian context, there is a systematic under-reporting of deaths attributable to heat because standardized coding through ICD-10, such as X30 (exposure to natural heat) and T67 (effects of heat and light), is rarely coded as the underlying cause of death, reflecting limited clinician recognition, a lack of protocols, and preference for proximate clinical conditions (e.g., circulatory or respiratory diseases) on death certificates. Given the multisystem effects of extreme heat, studies have often assessed heat-related mortality based on its contributing causes, particularly circulatory and respiratory diseases. Estimates of heat-attributable mortality, whether during HWs or inferred from associated causes, are highly sensitive to model specifications but lack specificity, which limits their accuracy in capturing the true magnitude of climate-related impacts on mortality.

Models such as distributed lag nonlinear models, time series with Poisson regression, and excess mortality estimates have been widely used in recent studies [39,40,41]. These approaches produce inferential estimates based on associated causes rather than on direct and specific clinical records identifying heat as a cause of death. This implies a shift in the produced knowledge, where we come to understand heat-related mortality not as an observable fact in health systems but as a phenomenon modeled from statistical patterns and frequently from aggregated environmental data. These models generally assume a nonlinear exposure–response function between temperature and mortality, with controls for long-term trends and seasonality, which may influence results, depending on the adopted specification.

Although necessary, this form of knowledge has both epistemological and practical implications. It enables the quantification of otherwise invisible impacts, highlights socio-environmental inequalities, and supports the development of public mitigation and adaptation policy initiatives. Among the strengths of statistical inference are its ability to capture the “hidden” burden of heat, to account for synergistic interactions with social inequalities and environmental stressors such as air pollution, and to provide evidence for early-warning systems and public health preparedness [42].

At the same time, its limitations are considerable. Estimates are highly sensitive to model specifications, including choices of lag structure and temperature thresholds, and depend heavily on the quality and spatial resolution of mortality and climate data. In Brazil, where access to timely and disaggregated health records is limited, these constraints increase the risk of overlooking vulnerable subpopulations, such as Indigenous peoples, informal workers, or residents of remote areas. Urban populations in informal settlements may also be missed due to incomplete mortality records. These data gaps reflect structural barriers that undermine the equity of public health responses. The absence of standardized definitions of HWs across studies also contributes to heterogeneity and hampers comparability. Excess mortality models may also be biased by confounding, particularly from air pollution, which may occur simultaneously with heat. Even advanced models that allow for covariates and interactive effects (e.g., heat with air pollution or humidity) rely on the availability of high-resolution, reliable datasets [42].

Furthermore, the reliance on estimates to support public health actions highlights a structural asymmetry: while other causes of death (such as heart attack, stroke, or pneumonia) are recorded individually, the impact of heat remains unrecorded on death certificates, making it difficult for the government to allocate resources, recognize the risk, and implement alerts and protective measures for vulnerable populations. Although estimates can guide effective interventions, particularly when targeting high-risk groups, this approach does not ensure that all vulnerable populations are identified. Some may remain overlooked, thereby undermining the reach and equity of public health responses.

To confront this invisibility, Brazil, in particular, needs stronger policy action to improve certification and coding practices (such as the systematic use of ICD-10 codes) and to better integrate health and climate information systems. Advancing these measures would provide timely and standardized data to support rapid decision-making and more effective responses to extreme heat. While the urgency is clear in the Brazilian context, these recommendations also carry relevance for other middle-income countries facing similar challenges of social inequality, informal labor, and vulnerable Indigenous or remote populations.

## 5. Building Resilience to Extreme Heat in Brazil

Mitigating the health impacts of HWs in Brazil requires more than technical measures; it demands the confrontation of structural weaknesses in governance, data, and equity. Preparedness and response plans are often cited as key solutions, but in practice they remain limited in scope and poorly adapted to Brazil’s context. These limitations reflect both the fragmentation of governance across sectors and the need for adaptive approaches that can respond to Brazil’s diverse social and territorial contexts.

Strategies must prioritize biologically and socially vulnerable populations but also ensure consistent implementation across regions. Mass gatherings, outdoor labor in agriculture and construction, and the millions of informal workers remain largely unprotected. Temporary cooling and hydration centers, workplace heat stress standards, and expanded access to cooling and transportation for low-income groups are frequently discussed but rarely institutionalized as health policy [43,44]. Recent initiatives illustrate both progress and limits. In 2024, Rio de Janeiro launched a municipal heat protocol, with risk levels linked to public alerts and emergency actions [39]. At the occupational level, the “MONITOR IBUTG app” was introduced to estimate WBGT exposure for outdoor workers [45]. These remain punctual initiatives, but they should be further explored to strengthen intersectoral action and heat governance at multiple levels.

Urban planning can play a significant role in mitigating urban heat island effects. The installation of cool roofs, green roofs, and cool pavement can help reduce surface temperatures in urban areas and provide natural cooling. Reflective “cool” roofs and cool pavement use coatings or light-colored materials that reflect solar radiation and reduce heat absorption, thereby mitigating urban heat island intensity. Modeling studies indicate they can lower local air temperatures by up to 1–2 °C, with co-benefits of reduced cooling demand [46]. Urban forests, parks, and protected green areas also play a critical role by providing shade and enhancing evapotranspiration, which reduces local heat stress and delivers additional ecosystem services that improve urban quality of life, e.g., improved air quality, stormwater regulation, and biodiversity conservation.

Despite their recognized benefits, the implementation of these measures in urban infrastructure planning in Brazil remains inconsistent and constrained by the availability of technical, institutional, and financial capacities at the local level. As a result, essential public facilities, such as healthcare units, schools, and shelters, are rarely integrated into broader resilience strategies, revealing a lack of effective cross-sectoral integration. Recent evidence also suggests that satellite-derived maps of urban heat islands, updated in near-real time, could support more targeted public health advisories by identifying hotspots and guiding the deployment of temporary cooling resources to the most vulnerable neighborhoods during HWs [47]. While such strategies illustrate urban responses, very different challenges are faced in remote and rural regions, where limited services and structural inequalities create distinct vulnerabilities.

Remote and vulnerable communities in the Amazon, including small towns, Indigenous peoples, and riverine populations, face limited access to specialized healthcare services, qualified professionals, financial resources, and political visibility, often isolated and lacking the infrastructure necessary to protect their populations from extreme heat [48,49]. Cultural specificities and logistical barriers, such as distance, transportation constraints, and language, further hinder adaptation measures [49,50]. Droughts and prolonged HWs, which reduce river flows and restrict water availability, including for subsistence, also force adjustments in traditional agricultural practices. These include coping with soil aridity, changes in working hours, or even reduced working hours, in addition to making changes in planting and opting for elements that easily adapt to divergent climate scenarios [51,52]. Such adaptations, however, require greater coordination and prioritization of public policies to guarantee consistent institutional support tailored to the specificities of these territories, addressing structural inequalities that have been historically neglected.

Implementing emergency response measures during periods of extreme heat, along with long-term adaptation strategies, is essential for reducing heat-related mortality and illness. This effort depends on the availability of accurate, timely, and sensitive data on the health effects of heat. It is necessary to monitor not only deaths but also adverse health outcomes and morbidity, including dehydration, hyperthermia, and exacerbation of pre-existing conditions, to enable more targeted and effective responses. Although Brazil has a national hospitalization database that is frequently used as a proxy to track such impacts, its administrative nature implies operational limitations, coverage restricted to users of the public health system, and delays in data availability. These shortcomings hinder timely action and underscore the fragility of relying exclusively on administrative data. The use of traditional health data could also strengthen surveillance. Primary care records and emergency room visits that do not result in hospitalization or death are still largely administrative in Brazil, but they could be standardized, harmonized, and integrated as complementary indicators of heat-related impacts. Together with digital sources (emergency call records, Google search trends, or social media), these data would allow for a more comprehensive and timely understanding of the health risks of heat. It is therefore crucial to move beyond dependence on these sources, advancing towards the harmonization of information already collected by different sectors and the development of new, more agile, and responsive monitoring systems.

In this context, Brazil must make progress not only in generating more sensitive and specific health data but also in adopting consistent definitions of HWs and developing context-appropriate metrics and indices that are timely, locally relevant, and accessible to the health sector. Event-based surveillance strategies can make a valuable contribution by enhancing the detection of heat-related health events through alternative and unstructured sources of information, such as syndromic reports, community observations, and media coverage. Modern computational tools like machine learning and natural language processing can help identify early signals of morbidity and mortality, even when specific datasets are lacking [53]. Experiences from air pollution monitoring show that integrating environmental and health indicators enables early warnings and informed responses while also providing co-benefits by reducing risks from overlapping exposures such as heat and air pollution. These approaches remain incipient in Brazil and will require political will, investment, and cross-sectoral collaboration to strengthen preparedness during extreme events.

At the same time, Brazil’s public health system provides extensive territorial coverage, reaching urban peripheries and some remote areas through its primary care and surveillance network. This capillarity is a strategic strength for heatwave monitoring and local response, but systemic challenges persist. The health sector lacks standardized protocols for recording heat-related morbidity and mortality, surveillance does not integrate specific early warning systems, and cross-sectoral fragmentation hampers coordinated response. Although the National Adaptation Plan includes health as a strategic axis, implementation remains uneven, particularly in vulnerable municipalities with limited resources and staff. Therefore, building resilience requires translation of national guidelines into effective local practices by strengthening primary care, integrating surveillance and assistance, developing intersectoral contingency plans, and investing in training and public engagement.

## 6. Conclusions

The challenge of adapting Brazil to HW events does not stem from a lack of knowledge or technological solutions or strategies but from the need to tailor strategies to Brazil’s diverse realities through a multisectoral and integrated approach. Brazil’s vast territory is characterized by profound social vulnerabilities, requiring adaptive frameworks that address its diverse social and geographic contexts. Achieving resilience and equitable adaptation therefore depends on context-specific approaches and strong political engagement, grounded in sustained intersectoral coordination and the active participation of local communities.

In conclusion, heat governance in Brazil is strongly place-based, relying on the capacity of local health systems to absorb and implement measures such as surveillance, early warning, and emergency care. At the same time, standardized protocols, integration of health and climate data, and cross-sectoral coordination can be generalized and scaled nationally to provide a common framework for adaptation and resilience. Brazil’s experience also offers valuable lessons for other middle-income countries with large Indigenous populations or extensive informal labor sectors, showing that a universal public health system with territorial reach, equity, and federative coordination can protect vulnerable groups and strengthen resilience to heat.

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
