# Peer review of "Too Hot to Ignore: The Escalating Health Impact of Heatwaves in Brazil"

_ijerph, 2025, doi:10.3390/ijerph22091451_

Round 1

Reviewer 1 Report

Comments and Suggestions for Authors

This opinion paper evaluates heat wave resilience in Brazil, arguing for systemic reform in Brazil’s public health apparatus. The paper does not explicitly identify a knowledge gap but does seem to respond to increasing incidence of heat waves in Brazil over the past two decades, including events in 2024 that combined extreme heat with air pollution from forest fires. The paper’s key contribution lies in identifying the intersecting causes of worsening heat-health vulnerability in Brazil and discussing governance approaches to building heat resilience. In what follows, I pose a series of questions for the authors as provocations for thought on how to sharpen this contribution. Responses to each should be integrated into any revision.

The paper is well-organized and grounded in the literature, incorporating numerous recent studies. However, the summary of existing knowledge tends to overshadow the key contribution, which is the opinion piece. To highlight this key contribution, the paper should explicitly state a knowledge gap it seeks to fill. It should also include a figure or table that enumerates the various dimensions of heat resilience in Brazil, connecting current problems with potential solutions and additional considerations, limitations, etc.. It would also be helpful for the conclusion to discuss how much heat governance, like the definition of a heat wave, is place-based and which aspects can be generalized.

Question: How can Brazil’s experience inform heat resilience in other countries, particularly other middle-income countries or countries with large indigenous populations or informal sectors?

The strongest policy recommendations and most interesting discussion emerge in Section 4, where the paper explains how current approaches to clinical records administration drive neglect of heat-health vulnerability by rendering heat-related mortality “invisible”. The paper explains how shifting from enumeration of cause of death to statistical inference can capture heat-health issues that emerge from synergistic interaction of heat with existing health issues, social inequality, and environmental change. The paper vaguely gestures towards various limits to inferential statistics, including sensitivity to model parameters, overlooked populations, and the need for high resolution data. More details are needed here to expand on the strengths and limitations of statistical inference.

Question: What policy recommendations do you have for Brazil and other countries in terms of improving heat related mortality record-keeping and making “visible” heat-health risks while also addressing the limitations of statistical inference?

The conclusion provides two additional insights into weak heat-health governance: fragmentation and adaptation. Information on both should be introduced earlier in the paper to set up this conclusion.

Question: In what ways is Brazil’s heat-health governance fragmented and how does an adaptive framework address Brazil’s unique heat-health risks in a more systematic fashion? Are there any case studies of good heat-health governance in Brazil or comparable country that can serve as a model for reform?

Specific comments

Abstract: Revise to clarify knowledge gap and key contribution.

Line 66: Briefly define public Unified Health System (SUS) and explain in 1-2 sentences what services are offered and to whom.

Lines 122-131: Given the topic of this special issue, a few more sentences on synergistic health impacts from air pollution and heat from the 2024 heat wave are warranted. This could help transition to the discussion of clinical records in Section 4.

Line 138-140: Rephrase this sentence, unclear.

Line 166-167: The authors state their recommendation of statistical estimates of heat wave mortality may overlook certain vulnerable populations. Expand this with a few sentences explaining who might be overlooked. Are there inherent data-related limits to equitable public health response to heat? If so, what are they?

Line 220-221: Explain what a “context-specific approach[h] and strong political engagement” means.

Author Response

Comments from Reviewer 1

General Comments:‎ This opinion paper evaluates heat wave resilience in Brazil, arguing for systemic reform in Brazil’s public health apparatus. The paper does not explicitly identify a knowledge gap but does seem to respond to increasing incidence of heat waves in Brazil over the past two decades, including events in 2024 that combined extreme heat with air pollution from forest fires. The paper’s key contribution lies in identifying the intersecting causes of worsening heat-health vulnerability in Brazil and discussing governance approaches to building heat resilience. In what follows, I pose a series of questions for the authors as provocations for thought on how to sharpen this contribution. Responses to each should be integrated into any revision.

The paper is well-organized and grounded in literature, incorporating numerous recent studies. However, the summary of existing knowledge tends to overshadow the key contribution, which is the opinion piece. To highlight this key contribution, the paper should explicitly state a knowledge gap it seeks to fill. It should also include a figure or table that enumerates the various dimensions of heat resilience in Brazil, connecting current problems with potential solutions and additional considerations, limitations, etc. It would also be helpful for the conclusion to discuss how much heat governance, like the definition of a heat wave, is place-based and which aspects can be generalized.

Response: We thank the reviewer for the constructive feedback and for highlighting key points to strengthen the contribution of our opinion paper. Below we provide responses to each of the main comments. Specific revisions were made to the introduction, Section 5, and the conclusion to address these concerns.

Comment 1: “To highlight this key contribution, the paper should explicitly state a knowledge gap it seeks to fill”: To address this comment, we added a paragraph at the end of the introduction explicitly stating the knowledge gaps (line 64 – 70). These include: (i) the invisibility of heat-related mortality in official health statistics, (ii) fragmented governance across sectors, and (iii) insufficient responses to compounded risks involving extreme heat, social inequalities, and environmental stressors.

Commnet 2: It should also include a figure or table that enumerates the various dimensions of heat resilience in Brazil, connecting current problems with potential solutions and additional considerations, limitations, etc”We appreciate this suggestion. Instead of adding a figure or table, we incorporated a narrative synthesis in Section 5 (Building Resilience in Brazil to Extreme Heat) that enumerates the main dimensions of resilience in Brazil, connecting current problems with potential responses and their limitations. This approach maintains the opinion style of the paper while addressing the reviewer’s concern.

Comment 3: “It would also be helpful for the conclusion to discuss how much heat governance, like the definition of a heat wave, is place-based and which aspects can be generalized”: We addressed this by adding a paragraph in the conclusion that highlights how heat governance in Brazil is strongly place-based, depending on local health system capacity, while also noting that elements such as standardized protocols, integration of health and climate data, and cross-sectoral coordination can be generalized and scaled nationally.

Question: How can Brazil’s experience inform heat resilience in other countries, particularly other middle-income countries or countries with large indigenous populations or informal sectors?

Response: We added to the conclusion that Brazil’s universal, equity-oriented public health system offers transferable lessons for other middle-income countries. Its experience shows how federative coordination can strengthen resilience in vulnerable contexts.

The strongest policy recommendations and most interesting discussion emerge in Section 4, where the paper explains how current approaches to clinical records administration drive neglect of heat-health vulnerability by rendering heat-related mortality “invisible”. The paper explains how shifting from enumeration of cause of death to statistical inference can capture heat-health issues that emerge from synergistic interaction of heat with existing health issues, social inequality, and environmental change. The paper vaguely gestures towards various limits to inferential statistics, including sensitivity to model parameters, overlooked populations, and the need for high resolution data. More details are needed here to expand on the strengths and limitations of statistical inference.

Response: We revised Section 4 by expanding the discussion of the limitations of statistical inference and included recent evidence from Graffy et al., 2024 to strengthen this point.

Question: What policy recommendations do you have for Brazil and other countries in terms of improving heat related mortality record-keeping and making “visible” heat-health risks while also addressing the limitations of statistical inference?

Response: We added a final paragraph in Section 4 highlighting the need to strengthen certification, coding, and data integration in Brazil, with lessons that also apply to other middle-income countries facing similar vulnerabilities.

The conclusion provides two additional insights into weak heat-health governance: fragmentation and adaptation. Information on both should be introduced earlier in the paper to set up this conclusion.

Response: We revised the opening of Section 5 to explicitly include governance fragmentation and the need for adaptive approaches, so these themes are introduced earlier and prepare the conclusion.

Question: In what ways is Brazil’s heat-health governance fragmented and how does an adaptive framework address Brazil’s unique heat-health risks in a more systematic fashion? Are there any case studies of good heat-health governance in Brazil or comparable country that can serve as a model for reform?

Response: We revised the opening of Section 5 to explicitly include governance fragmentation and the need for adaptive approaches, so these themes are introduced earlier and prepare the conclusion.

Specific comments

Abstract: Revise to clarify knowledge gap and key contribution.

Response: We revised the abstract to clearly state the knowledge gap and our main contribution

Line 66: Briefly define public Unified Health System (SUS) and explain in 1-2 sentences what services are offered and to whom.

Response: We thank the reviewer for this observation. A brief explanation of the Unified Health System (SUS) has been added.

Lines 122-131: Given the topic of this special issue, a few more sentences on synergistic health impacts from air pollution and heat from the 2024 heat wave are warranted. This could help transition to the discussion of clinical records in Section 4.

Response: At the end of Section 3, we included a discussion of the synergistic health impacts of extreme heat and air pollution during the 2024 heat wave.

Line 138-140: Rephrase this sentence, unclear.

Response: The sentence was revised to clarify sensitivity to model choices and limited specificity in estimating heat-attributable mortality.

Line 166-167: The authors state their recommendation of statistical estimates of heat wave mortality may overlook certain vulnerable populations. Expand this with a few sentences explaining who might be overlooked. Are there inherent data-related limits to equitable public health response to heat? If so, what are they?

Response: We revised the section to specify which vulnerable groups may be overlooked in statistical estimates, including Indigenous peoples, informal workers, residents of remote areas, and urban populations in informal settlements

Line 220-221: Explain what a “context-specific approach[h] and strong political engagement” means.

Response: We clarified that context-specific approaches adapt strategies to regional and social realities, while strong political engagement involves intersectoral coordination and local participation.

Reviewer 2 Report

Comments and Suggestions for Authors

See attachment.

Author Response

Comments from Reviewer 2

Review of “Too hot to ignore: the escalating health impact of heatwaves in Brazil” The strength of the manuscript lies in its effective integration of public health, socio- economic, and environmental dimensions. However, several key areas require significant revision to elevate the manuscript's contribution and scientific rigor:

Major issues:

  1. The manuscript correctly states that no universally accepted definition for a HW exists and provides a common example (Tmax> average Tmax by 5°C for 3+ days). This point is critical and deserves much deeper exploration. Brazil has immense climatic diversity, from the humid tropics of the Amazon to the semi-arid Sertão andtemperate south.

A single, temperature-anomaly-based definition is insufficient. The authors should discuss the importance of regionally calibrated, multivariate indices that incorporate humidity (e.g., heat index, wet-bulb globe temperature) and are tied to local acclimatization levels. The lack of such a standard poses a major barrier to effective public health warnings and comparative research, a point the paper should elaborate on. While,the paper mentions El Niño as a factor in the 2023/2024 heat. This could be expanded to briefly discuss other key atmospheric drivers of HWs in Brazil, such as atmospheric blocking patterns or sea surface temperature anomalies in the Atlantic, to provide a richer meteorological context.

Response: To address this comment, we expanded the Context section to highlight Brazil’s climatic diversity and the limitations of a single temperature-anomaly-based definition. We now discuss the importance of regionally calibrated and multivariate indices, and added meteorological drivers beyond El Niño, such as atmospheric blocking and Atlantic SST anomalies, to better contextualize the 2023/2024 heatwaves. We also included an additional reference (Marengo et al., 2025) to support this discussion.

  1. The solutions proposed in Section 5 are generally sound but largely generic (e.g., "cooling centers," "planting trees"). The recommendations would be more impactful if they were more specific to the challenges previously identified. For example, the paper calls for "more sensitive and specific health data"10. What does this mean in practice? The authors could suggest leveraging non-traditional data sources like syndromic surveillance from emergency calls, Google search trends, or social media data, analyzed via natural language processing, to achieve the "real- time response" the paper advocates for. On the other hand, how can urban planning initiatives be integrated with public health advisories? For example, how can real- time urban heat island maps, perhaps derived from satellite data, be used to deploy mobile cooling resources to the most vulnerable neighborhoods during a HW? The recommendations should be more concrete and technologically informed.

Response: We revised Section 5 to make the recommendations more specific and technologically informed. In practice, we included the potential use of complementary health data (e.g., primary care and emergency visits) together with non-traditional sources such as emergency calls, Google search trends, and social media to improve real-time surveillance. We also highlighted how satellite-derived urban heat island maps can be used to guide mobile cooling resources to the most vulnerable neighborhoods during heatwaves (Zhao et al., 2025).

Minor issues:

  1. Line 30: The definition provided ("[...] daily maximum temperature exceeding the average maximum temperature by at least 5∘C [1]") is crucial to specify the baseline period for the "average maximum temperature" (e.g., 1991-2020), as this choice significantly impacts HW detection.

Response: The definition of heatwaves was revised to align with established references, including WHO/WMO guidance (McGregor et al., 2015) and Robinson (2001). We also specified the use of a 30-year climatological baseline for anomaly-based thresholds to ensure greater clarity and comparability across contexts.

  1. L. 39: The citation for the 2023/2024 summer heat record ([4]) has a future publication date ("Publicado Em 20/03/2025") and appears to refer to the "Verão 2024-2025" in its title. This must be corrected. Please verify the source and link.

Response: The citation for the 2023/2024 summer heat record has been corrected. We verified the source and updated the reference to ensure consistency with the appropriate publication and period.

  1. L. 128: When discussing the synergistic effects of heat and fine particulate matter (PM2.5), consider briefly mentioning the source of these pollutants (e.g., biomass burning from deforestation and agricultural practices), as this is a major issue in the Amazon and other parts of Brazil.

Response: We thank the reviewer for this suggestion. We have incorporated a brief mention of deforestation and agricultural burning as major sources of wildfire-related air pollution in the Amazon, without overloading the text, to highlight the relevance of these drivers in Brazil.

  1. L. 208: The mention of "event-based surveillance strategies" is good, However, as noted in the major comments, the authors should expand on this with specific examples of how this can be implemented using modern computational tools.

Response: We appreciate this observation. We revised the section to note that modern computational tools can help detect heat-related morbidity and mortality, particularly where access to health data is limited. We also emphasized integrated approaches, such as those used in air pollution monitoring, which provide co-benefits by addressing overlapping exposures like heat and air pollution.

Reviewer 3 Report

Comments and Suggestions for Authors

Overall assessment:

The paper provides a comprehensive examination of the escalating health impacts of heatwaves in Brazil, addressing physiological effects, vulnerable populations, and societal consequences. The research is well-structured and relevant, but several areas require refinement, including methodological transparency, data clarity, policy feasibility, and language precision. Once these revisions are made, the manuscript may be more suitable for publication in the Int. J. Environ. Res. Public Health.

Major comments:

  1. The statistical models (e.g., distributed lag nonlinear models) and analytical approaches need more detailed descriptions, including parameter selections, assumptions, and limitations.
  2. The paper cites increased heatwaves in Brazilian capitals but lacks granularity (e.g., regional variability, urban vs. rural disparities). More localized data would strengthen the analysis.
  3. Proposed strategies (e.g., cooling centers, urban greening) are generic. Tailored, actionable steps—such as cost estimates, implementation timelines, or case studies—would enhance practicality.
  4. While briefly mentioned, the unique challenges faced by Amazonian and indigenous communities deserve deeper exploration, including cultural and logistical barriers to adaptation.

Minor Comments:

  1. Abstract: Replace "comprehensive article" with "systematic review" or "analytical study" for precision.
  2. Section 1: Clarify the definition of heatwaves (HWs) by citing Brazil-specific thresholds if available.
  3. Line 71 : Define "hyperthermia" for non-medical readers.
  4. Line 99: "Neoplasms" should be replaced with "cancer-related deaths" for clarity.
  5. Line 111: Provide Brazilian examples of heat-linked violence (e.g., specific metropolitan studies).
  6. Section 4: Explain why ICD-10 codes (X30/T67) are underused in Brazil (e.g., clinician awareness gaps).
  7. Line 156: Acknowledge potential biases in excess mortality models (e.g., confounding by air pollution).
  8. Section 5: "Cool roofs/pavements" need technical details (e.g., materials, cost-benefit comparisons).
  9. Address how Brazil’s fragmented governance might hinder cross-sectoral coordination.
  10. Figures:Include a map of Brazil highlighting HW frequency by region.
  11. Tables: Add a table summarizing heat-attributable mortality estimates from cited studies.
  12. Avoid passive voice (e.g., "It has been shown" → "Studies demonstrate").

Author Response

Comments from Reviewer 3

The paper provides a comprehensive examination of the escalating health impacts of heatwaves in Brazil, addressing physiological effects, vulnerable populations, and societal consequences. The research is well-structured and relevant, but several areas require refinement, including methodological transparency, data clarity, policy feasibility, and language precision. Once these revisions are made, the manuscript may be more suitable for publication in the Int. J. Environ. Res. Public Health.

 Major comments:

  1. The statistical models (e.g., distributed lag nonlinear models) and analytical approaches need more detailed descriptions, including parameter selections, assumptions, and limitations.

Response: We thank the reviewer for this comment. We added details on model assumptions, parameter choices, and limitations, and reinforced the discussion with the inclusion of Graffy et al. (2024).

  1. The paper cites increased heatwaves in Brazilian capitals but lacks granularity (e.g., regional variability, urban vs. rural disparities). More localized data would strengthen the analysis.

Response: In the revised version, we broadened the scope in Section 5 by discussing regionally specific challenges for the Brazilian health system, including contrasts between urban areas and remote or rural communities such as small towns, Indigenous peoples, and riverine populations in the Amazon.

  1. Proposed strategies (e.g., cooling centers, urban greening) are generic. Tailored, actionable steps—such as cost estimates, implementation timelines, or case studies—would enhance practicality.

Response: We addressed this point by adding concrete actions in Section 5, including cool roofs, urban greening, and satellite-based heat maps. We also highlighted Brazilian initiatives such as the Rio de Janeiro heat protocol and the MONITOR IBUTG app. While we do not provide cost estimates or timelines, these examples illustrate practical steps and limitations.

  1. While briefly mentioned, the unique challenges faced by Amazonian and indigenous communities deserve deeper exploration, including cultural and logistical barriers to adaptation.

Response: We incorporated this point in Section 5 by adding a discussion of cultural and logistical barriers faced by Amazonian and Indigenous communities. We also included a reference to Garnelo (2019), which highlights the specificities and challenges of public health policies in the Brazilian Amazon, to strengthen this argument

Minor Comments:

1.Abstract: Replace "comprehensive article" with "systematic review" or "analytical study" for precision.

Response: We agree that “comprehensive article” lacked precision. Since this is not a systematic review or analytical study, we revised the abstract to describe it as a “perspective article,” which better reflects the scope and objectives of the paper.

2.Section 1: Clarify the definition of heatwaves (HWs) by citing Brazil-specific thresholds if available.

Response: In Section 1, we incorporated Brazil-specific definitions of heatwaves. Evidence from Rio de Janeiro and the Amazon shows that thresholds based on the 90th–99th percentiles of daily mean temperature sustained for at least two days are associated with higher mortality risks, particularly among older adults and women [Silveira et al., 2023a; Silveira et al., 2023b].

3.Line 71 : Define "hyperthermia" for non-medical readers.

Response: We added a brief definition of hyperthermia for clarity.

4.Line 99: "Neoplasms" should be replaced with "cancer-related deaths" for clarity.

Response: We replaced “neoplasms” with “cancer-related deaths” for clarity.

5.Line 111: Provide Brazilian examples of heat-linked violence (e.g., specific metropolitan studies).

Response: In the revised version, national evidence was incorporated into the text for a broader perspective.

6.Section 4: Explain why ICD-10 codes (X30/T67) are underused in Brazil (e.g., clinician awareness gaps).

Response: We revised the text to clarify the reasons for the underuse of ICD-10 codes (X30/T67) in Brazil, highlighting limited clinician recognition, absent protocols, and the tendency to record proximate clinical conditions on death certificates.

  1. Line 156: Acknowledge potential biases in excess mortality models (e.g., confounding by air pollution).

Response: Thank you for the suggestion. We have revised the text to explicitly acknowledge potential biases in excess mortality models, particularly confounded by air pollution, which may occur simultaneously with heat.

8.Section 5: "Cool roofs/pavements" need technical details (e.g., materials, cost-benefit comparisons).

Response: We have revised the section to include a reference (Macintyre & Heaviside, 2019) that highlights technical details and comparative benefits of reflective “cool” roofs and pavements. This allows us to address the reviewer’s point without adding excessive detail that could make the text too dense.

  1. Address how Brazil’s fragmented governance might hinder cross-sectoral coordination.

Response: This aspect is already addressed in Section 5 and further reinforced in the Conclusions.

Figures:Include a map of Brazil highlighting HW frequency by region.

Tables: Add a table summarizing heat-attributable mortality estimates from cited studies.

Response: We thank the reviewer for this constructive suggestion. As an opinion article, our focus is on synthesizing evidence and highlighting policy and governance challenges rather than providing detailed quantitative analyses, which would be more appropriate for a systematic review. For this reason, we did not include tables or maps with disaggregated estimates, though we would be glad to add a simple illustrative map if the editor considers it useful.

Avoid passive voice (e.g., "It has been shown" → "Studies demonstrate").

Response: We thank the reviewer for this observation. The manuscript had already been reviewed in its first version and was subsequently revised by a native speaker following the reviewers’ comments.

Round 2

Reviewer 2 Report

Comments and Suggestions for Authors

I have no further comments.